# The Lung Microbiome and Its Impact on Obstructive Sleep Apnea: A Diagnostic Frontier

**DOI:** 10.3390/diagnostics15111431

**Published:** 2025-06-04

**Authors:** Aliki Karkala, Serafeim-Chrysovalantis Kotoulas, Asterios Tzinas, Eleni Massa, Eleni Mouloudi, Foteini Gkakou, Athanasia Pataka

**Affiliations:** 1Respiratory Failure Unit, General Hospital of Thessaloniki “G. Papanikolaou”, Exochi, 57010 Thessaloniki, Greece; stergiostzinas@hotmail.com (A.T.); patakath@yahoo.gr (A.P.); 2Intensive Care Unit, “Hippokration” Hospital of Thessaloniki, 54642 Thessaloniki, Greece; elenizioga@yahoo.com (E.M.); elmoulou@yahoo.gr (E.M.); 3Department of Pulmonology, General Hospital of Thessaloniki “G. Papanikolaou”, Exochi, 57010 Thessaloniki, Greece; fgkakou@hotmail.com

**Keywords:** dysbiosis, lower respiratory tract, lung microbiome, obstructive sleep apnea

## Abstract

Obstructive sleep apnea (OSA), a prevalent disorder characterized by recurrent upper airway collapse, is increasingly recognized as a systemic inflammatory condition influenced by microbial dysregulation. Emerging evidence underscores the lung microbiome as a mediator in OSA pathophysiology, where dysbiotic shifts driven by intermittent hypoxia, oxidative stress and mechanical airway trauma amplify inflammatory cascades and perpetuate respiratory instability. This review synthesizes current knowledge on the bidirectional interplay between OSA and lung microbial communities. It aims to highlight how hypoxia-induced alterations in microbial ecology disrupt immune homeostasis, while inflammation-driven mucosal injury fosters pathogenic colonization. Clinical correlations between specific taxa like *Streptococcus* and *Prevotella*, and disease severity, suggest microbial signatures as novel biomarkers for OSA progression and treatment response. Furthermore, oxidative stress markers and pro-inflammatory cytokines emerge as potential diagnostic tools that bridge microbial dysbiosis with sleep-related outcomes. However, challenges persist in sampling standardization of the low-biomass lower airways, as well as in causative mechanisms linking microbial dysbiosis to OSA pathophysiology. By integrating microbial ecology with precision sleep medicine, this paradigm shift promises to transform OSA management from mechanical stabilization to holistic ecosystem restoration.

## 1. Introduction

The human lung is recognized as a dynamic ecosystem that harbors diverse microbial communities with pivotal roles in respiratory health and disease [1]. Although less densely populated than the gastrointestinal tract, the lung microbiome is composed of the bacteriome, mycobiome and virome. Among the bacteriome, *Streptococcus*, *Veillonella* and *Prevotella* are the most common genera, while *Haemophilus* species are unique to the lung as resident inhabitants and are rare in other microbiomes. The mycobiome is less numerous, with *Candida* dominating, followed by *Saccharomyces* and *Penicillium*. *Phages* dominate the virome alongside a small number of respiratory viruses [2]. This community appears to be shaped by a delicate equilibrium between microbial immigration from the upper respiratory tract (URT), local elimination via host defenses and microenvironmental factors such as oxygen tension, pH and nutrient availability [3,4]. Despite its low biomass, the lung microbiome regulates epithelial barrier integrity, resistance to pathogens and inflammatory responses through immunomodulatory effects [5]. Lung dysbiosis, the disruption of microbial equilibrium, has been implicated in a variety of chronic respiratory conditions [6,7] but its role in sleep-related disorders, particularly obstructive sleep apnea (OSA), remains underexplored despite compelling mechanistic overlaps. OSA is characterized by recurrent upper airway collapse during sleep, leading to intermittent hypoxia (IH), sleep fragmentation and systemic inflammation [8]. It represents a disorder of rising global prevalence with immediate morbidity such as excessive daytime sleepiness and links to cardiovascular disease and metabolic syndrome, incurring substantial socioeconomic costs [9]. Furthermore, despite advances in continuous positive airway pressure (CPAP) therapy, the gold standard for OSA management, suboptimal adherence (>50% non-compliance) and the limitations of apnea-hypopnea index (AHI)-centric diagnostics underscore the need for precision medicine approaches that address its multifactorial pathogenesis [10].

While gaining popularity, research on how lung microbiota contributes to OSA is still in its infancy. OSA-induced hypoxia, mechanical stress from airway collapse and systemic inflammation create a microenvironment conducive to microbial dysbiosis, favoring anaerobic pathobionts while depleting commensal taxa [11,12]. Conversely, dysbiosis may exacerbate OSA severity by amplifying mucosal inflammation, impairing neuromuscular reflexes and promoting airway instability [11,13]. However, the field remains hindered by methodological challenges in sampling low-biomass communities and a paucity of longitudinal data.

This narrative review aims to critically synthesize current evidence on the mechanistic interplay between lung microbiome dysbiosis and OSA pathogenesis. Specifically, it aims to (1) examine how core OSA pathophysiological features such as intermittent hypoxia, mechanical stress and systemic inflammation drive compositional and functional alterations in the lung microbiome; (2) explore the mechanisms by which resultant dysbiosis may perpetuate or exacerbate OSA severity through inflammatory cascades, immune dysregulation and impaired airway stability; (3) evaluate the potential of specific microbial signatures and associated inflammatory markers as biomarkers for OSA severity, progression, or therapeutic response. By integrating insights from microbial ecology, immunology and sleep medicine, this review seeks to advance the understanding of OSA as a disorder of disrupted host-microbe crosstalk and highlight the lung microbiome’s emerging role as a diagnostic and therapeutic target.

## 2. The Lung Microbiome in Health and Disease

### 2.1. Composition of the Lung Microbiome in Healthy Individuals

The lung microbiome, though less characterized than microbiomes of other body sites, plays a critical role in maintaining respiratory health. However, it remains understudied compared to other body sites, largely due to technical challenges in non-invasive sampling of the lower respiratory tract (LRT) and its overlap with URT microbiome [1]. However, bacterial density in the LRT is orders of magnitude lower than in the URT [3] with community composition closely mirroring that of the oropharynx and representing a significant biomass gradient, where transient perturbations can have extended effects on host inflammatory response [2]. While the nasal cavity microbiota is enriched with *Staphylococcus*, *Propionibacterium*, *Corynebacterium*, *Moraxella*, and *Streptococcus* [14,15], a microbial density niche is observed in the oropharynx, with the most representative species being *Streptococcus*, *Veillonella* and *Prevotella* [16]. The core bacterial microbiome of healthy lungs is dominated by the phyla *Firmicutes* and *Bacteroidetes*, with *Prevotella*, *Streptococcus,* and *Veillonella* representing the most abundant genera [17]. While most bacteria are aerobic or facultative anaerobic, specialized anaerobes such as *Clostridium*, *Veillonella* and *Porphyromonas* are also present [2].

Fungal communities in healthy lungs are diverse but insufficiently characterized, perhaps due to low biomass, difficulty in DNA extraction and inconsistencies in taxonomic classification [18]. Predominant taxa include *Ascomycetes* and *Basidiomycetes*, with *Candida*, *Saccharomyces*, *Penicillium,* and *Aspergillus* frequently detected [19,20,21]. Unlike bacteria, which often directly influence pulmonary homeostasis through active colonization and metabolite production, current evidence suggests fungi primarily function as immunomodulatory cofactors. This implies they do not typically initiate or dominantly regulate baseline pulmonary immune tone themselves. Instead, they modulate ongoing immune responses potentially initiated by bacteria, viruses, or other stimuli, through interactions with host pattern recognition receptors and the release of metabolites that can amplify or dampen inflammatory pathways [22]. Furthermore, interactions between fungi and bacteria, such as biofilm formation, enhance microbial resilience against host defenses and antimicrobial agents, potentially driving multidrug resistance [23]. Clinical interest in the mycobiome has centered on its dual role as both a symbiotic partner and a latent pathogen, with implications for disease prevention and therapeutic targeting [24].

Finally, the pulmonary virome, a community of low complexity in healthy individuals, comprises prokaryotic and eukaryotic viruses, albeit at lower abundance than the oropharynx, with predominant families including *Paramyxoviridae*, *Picornaviridae* and *Orthomyxoviridae* [25], alongside the recently identified *Redondoviridae* [26]. Evidence from a metagenomic analysis has established that healthy individuals exhibit reduced viral community complexity compared to diseased states [27]. Latent viral infections may confer immunological benefits by priming the host’s immune system against secondary bacterial infections [28,29] and especially *phages*, which modulate bacterial populations through prophage-mediated competition [30], are abundant and may form a stable “core” community in the respiratory tract [30,31]. Meanwhile, latent viruses paradoxically have been reported to enhance immune vigilance [28] by regulating bacterial diversity, while fungal communities, though less abundant, may amplify inflammatory pathways or protect against dysbiosis [32]. Hence, future multi-kingdom analyses could concentrate on the collaboration of viruses with other microorganisms in the lung microbiome, as it may unravel disease mechanisms by clarifying interkingdom interactions.

### 2.2. Potential Origin of Lung Microbiome

The oropharyngeal microbiome has been extensively studied [33] and is recognized for its stability and significant influence on host physiology and pathology. Comparative analyses have revealed partial taxonomic overlaps between the oral cavity and lung microbiomes [6,34], although with distinct compositional and density-related differences. Healthy lung microbial communities have demonstrated a closer phylogenetic relationship to oral microbiota than to nasal microbiota, which contributes little to the lung microbiome of healthy humans [3], perhaps due to the low volume of nasal secretions relative to salivary flow in health [4]. Evidence has suggested that the lung microbiome may originate from the URT, with the primary mechanism linking these anatomically contiguous sites being microaspiration [1,35,36]. During normal respiration and particularly during sleep, small volumes of oropharyngeal secretions are passively aspirated into the LRT (see Figure 1). This process introduces a continuous influx of microbial immigrants from the high-biomass oropharynx into the low-biomass lung environment [1,35]. Despite this continuity in the airway mucosa [37,38], pulmonary microbial populations are sparser and less diverse than those of the oropharynx, with bacterial levels diminishing distally and modest regional variations suggesting differential dynamic equilibrium model of clearance and limited local replication [2].

Early hypotheses positing equivalence between lung and oropharyngeal microbiomes have been refined, acknowledging selective pressures exerted by lung-specific clearance mechanisms (e.g., mucociliary action, immune responses) and surfactant-mediated inhibition [3,14,39,40]. This combination of immigration and elimination, with minimal local replication under healthy conditions, shapes a transient, low-biomass community distinct from its oropharyngeal source. The selective pressures exerted by lung-specific clearance mechanisms and the physicochemical environment mean the LRT microbiome is not simply a diluted copy of the oropharynx but a selectively filtered derivative [3,14,39,40]. However, disease states disrupt this equilibrium, and structural damage, impaired clearance through reduced mucociliary function, inflammation, or altered conditions like hypoxia in OSA, can favor microbial proliferation, overriding the immigration-clearance balance and allowing dysbiotic communities to establish [1].

Nevertheless, a consensus has emerged that the oropharyngeal microbiome critically influences the establishment and modulation of lung microbial communities, with oral health implicated in respiratory disease risk [2]. Consequently, OSA may increase both the volume and altered composition of aspirated material while simultaneously reducing the LRT’s capacity to clear or control it, creating a perfect storm for seeding and perpetuating lung dysbiosis (Figure 1). This establishes a direct pathophysiological pathway linking OSA to potential LRT microbial alterations via its impact on the oropharynx and clearance mechanisms. While exhibiting interindividual variability, URT has been observed to remain temporally consistent in the absence of external perturbations [41,42] possibly establishing it as a reliable biomarker for tracking pulmonary microbial communities.

Finally, it is worth mentioning that the gut microbiome, a complex consortium dominated by the phyla *Firmicutes*, *Bacteroidetes* and *Actinobacteria* [43,44], has been implicated in modulation of pulmonary immunity and microbial composition. Despite the distinctions between gut and lung microbiota, their shared embryonic origin, structural parallels in mucosal epithelia and synchronized microbial colonization in early life could suggest an evolutionarily conserved crosstalk [45,46]. Furthermore, accumulating evidence has underscored that gut-derived metabolites, microbial ligands and immune cells, traverse the bloodstream to shape both innate and adaptive immune responses and influence lung homeostasis [47]. For instance, gut-initiated signals were observed to alter circulating and pulmonary metabolites [48], amplifying oxidative stress and contributing to lung injury [49]. Collectively, these findings could position the gut microbiome as a pivotal regulator of pulmonary immunity, offering novel therapeutic avenues for respiratory disorders rooted in microbial dysbiosis [45,46]. However, the most frequently observed microorganisms in the lower airways are those characteristically present in the oral cavity.

However, controversy has persisted regarding the composition of pulmonary microbiota as it appears to be dynamically shaped by selective pressures stemming from host immune responses and physicochemical gradients that drive microbial community assembly but also introduces variability that complicates cross-study comparisons in respiratory microbiome research [5]. Persistent methodological heterogeneity such as divergent sampling protocols and contamination risks during bronchoscopic procedures has led to inconsistent findings in the low-biomass LRT [11]. DNA contaminants from sterile instruments or saline often obscure true microbial signals by skewing culture-dependent analyses and masking dysbiotic states [50]. Furthermore, low-abundance taxa, though underrepresented in relative abundance metrics, may exert disproportionate pathogenic effects as keystone species [51]. Disregarding such taxa risks misinterpretation of microbial roles in disease, underscoring the need for refined analytical frameworks to resolve ecological and functional contributions of minority populations.

### 2.3. Pathological Dysbiosis and Its Implications for Respiratory Inflammation and Immune Response

Dysbiosis in the respiratory tract—a deviation from the microbiota composition observed in healthy individuals—remains contentious due to the lung’s low microbial biomass and dynamic interactions with environmental and systemic factors. Unlike the gut, where dysbiosis is often defined by proportional shifts in dominant taxa, lung dysbiosis encompasses both altered microbial proportions and total bacterial load, complicating its characterization in lung diseases [1]. A balanced airway microbiota has been established as critical for respiratory health, impacting host resistance to pathogen colonization via nutrient competition, antimicrobial peptide secretion and immune regulation [15,52]. However, host-specific factors like genetics, immune status and increased entry rates like in aspiration, as well as environmental triggers like diet, pharmaceuticals and pollution, have been identified as disruptors to this equilibrium, promoting pathobiont overgrowth and epithelial barrier dysfunction, even in diseases not traditionally associated with microbial etiology [52,53,54,55,56]. For instance, mucus accumulation in chronic obstructive pulmonary disease or asthma has been documented to create a nutrient-rich environment that favors pathogens like *Streptococcus pneumoniae* or Gram-negative bacteria, which often precede pneumonia [36,57]. While implicated in numerous pulmonary diseases, including asthma [58] and even lung cancer [59], whether dysbiosis initiates and perpetuates inflammatory injury or merely reflects disease progression as a marker remains unresolved [1].

LRT microbiome has been implicated in the regulation of the immunological tone of the airway mucosa, potentially serving to prime the immune system or modulate inflammatory processes, by scavenging microbes, deploying alveolar macrophages for antigen presentation, and regulating cytokine and chemokine signaling to maintain immune homeostasis [11]. Dysbiosis of the lung microbiome has been evidenced to disrupt immune homeostasis through multifaceted mechanisms, impacting both innate and adaptive immunity, with depletion of commensal species and overgrowth of pathobionts destabilizing epithelial barrier integrity, facilitating microbial toxin leakage and reactive oxygen species (ROS) production, which activate pattern recognition receptors [5]. This cascade has been implicated in pro-inflammatory cytokine release and neutrophil recruitment, perpetuating tissue damage and chronic inflammation, as observed in murine models of Proteobacterium catarrhalis infection [1,60,61]. Concurrently, dysbiosis has been revealed to skew immune polarization; enrichment of oral commensals like *Veillonella* and *Prevotella* in the lungs has been correlated with elevated immune responses [13], exacerbating mucosal inflammation [62] and impairing lung function [63]. Paradoxically, increased airway microbial diversity that is often deemed protective, has been correlated with bronchial hyperresponsiveness in asthma, particularly with *Proteobacteria* enrichment, highlighting context-dependent microbial roles [62].

The adaptive immune system appears to be similarly perturbed, as a growing body of evidence has supported the notion that dysbiosis alters dendritic cell priming and T-cell differentiation that modulates pulmonary inflammation [53,54], with gut-lung axis interactions playing a critical role [64]. Conversely, lung dysbiosis has been involved in local adaptive immunity disruption; for instance, fungal-derived tryptophan metabolites have been shown to amplify regulatory T-cells activity while suppressing Th17-mediated inflammation [65], a balance critical for resolving airway hyperreactivity [58]. However, dysbiosis-induced imbalances in short-chain fatty acids (SCFAs) and other microbial metabolites can impair this regulation, fostering environments conducive to pathogen colonization, as seen in Staphylococcus-driven SCFA fluctuations [61,66]. Therapeutic interventions, such as azithromycin in emphysema, have illustrated the microbiome’s dual role as both a driver and a therapeutic target, and while reducing inflammation, such treatments may inadvertently enrich stress-related microbial metabolites [1]. Yet conflicting data on microbial diversity’s role—protective in some contexts, pathogenic in others—demand phenotype-specific strategies [61]. Ultimately, whether dysbiosis is a primary instigator or secondary marker of disease remains unresolved, necessitating longitudinal studies to disentangle causality and refine microbiota-targeted therapies. While advances in multi-omics have elucidated microbial-immune crosstalk, translating these insights into therapies requires mechanistic clarity and personalized approaches.

## 3. Microbiome Dysbiosis and OSA

### 3.1. Emerging Research on Microbial Profiles in OSA Patients

OSA is characterized by recurrent, partial or complete, upper airway collapse during sleep that results in fragmented sleep, reduced blood oxygen saturation and hypercapnia. It is a highly prevalent disorder, associated with increased morbidity, mortality and socioeconomic costs [8]. Recurrent airway obstruction in OSA patients results in disturbances in the URT microbiome, including physiological mucosal alterations and the development of local inflammation [67], underscoring the role of respiratory tract microbiome dysbiosis in the disease. However, little is known about the airway microbiome composition in patients with OSA [12] primarily due to the invasiveness and challenges of sampling the low-biomass LRT. Consequently, most available data are derived from studies of the upper airways (nasopharynx, oropharynx), which, as the primary source of lung microbes via microaspiration [1,35,36], provide crucial, mechanistically relevant proxies for inferring LRT alterations, given this anatomical and physiological continuum.

Severe OSA has been linked to an enrichment of *Streptococcus*, *Prevotella* and *Veillonella* genera within the nasal microbiome, alongside elevated systemic inflammatory markers [68]. Although no significant differences in microbial composition were observed between individuals with mild to moderate OSA and healthy controls, dysbiosis may still contribute to the release of pro-inflammatory cytokines, such as interleukin-6 (IL-6) and IL-8, potentially exacerbating inflammatory pathways [68]. Indeed, in a general population cohort where the prevalence of severe OSA was presumed to be low, no distinct nasopharyngeal microbiome signatures differentiated OSA patients from non-OSA individuals [69]. However, subsequent analysis revealed divergent microbial community structures between these groups, with a significant negative correlation identified between the relative proportion of *Finegoldia magna*—a Gram-positive facultative anaerobe capable of thriving under hypoxic conditions—in nasopharyngeal samples and nocturnal oxygen saturation levels [69]. This suggests that hypoxia-driven alterations in the airway microenvironment may selectively promote the proliferation of facultative anaerobes, thereby reshaping microbial ecology. Notably, *F. magna*, typically a human commensal, functions as an opportunistic pathogen with documented capacity to produce pro-inflammatory mediators [70,71]. Furthermore, microbial metabolic byproducts have been implicated in modulating local epithelial inflammation [72,73], which may perpetuate mucosal inflammation. Such inflammatory processes are critical in OSA pathophysiology, as they could impair upper airway neuromuscular responsiveness [74] and are increasingly recognized as central drivers of disease progression [68,75]. Thus, the enrichment of hypoxia-adapted anaerobes like F. magna may not only reflect environmental adaptations but also actively contribute to inflammatory cascades and OSA pathogenesis. These findings suggest that microbial dysregulation may interact with hypoxic conditions in OSA pathophysiology, warranting investigation into causal mechanisms.

While anaerobic or facultative anaerobic taxa have been shown to dominate in severe OSA, aligning with hypoxia-driven microbial selection [68], another study identified distinct alterations in less severe OSA [76]. *Xanthobacteraceae* abundance was elevated in moderate OSA compared to non-OSA subjects, alongside reduced proportions of *Burkholderiales incertae sedis* and *Lachnospira*; in severe OSA, *Xanthobacteraceae* enrichment persisted, with additional increases in *Actinomycetaceae* and *Peptoniphilaceae* [76]. While *Xanthobacteraceae* are transient gastrointestinal commensals [77], *Burkholderiales incertae sedis* and *Comamonadaceae* are recognized as typical nasal cavity symbionts [3]. As the latter dominate oxygen-rich environments [78], their displacement by anaerobic taxa in OSA may reflect reduced airway oxygen levels or biofilm-induced hypoxia [78]. Notably, the most abundant taxa aligned with healthy nasal microbiome profiles [79], and after 6–9 months of CPAP therapy, OSA patients’ microbial biodiversity shifted toward control-like patterns, suggesting a gradual normalization process [76]. Improved oxygenation due to CPAP-mediated airway patency may explain the resurgence of aerobic bacteria, counteracting hypoxia-driven anaerobe proliferation. These findings underscore the dynamic interplay between OSA severity, airway hypoxia and microbial ecology, with CPAP emerging as a potential modulator of microbiome restoration. However, temporal and anatomical factors critically influence these relationships, necessitating longitudinal, site-specific analyses to clarify causal mechanisms in OSA pathogenesis.

In the oropharynx, OSA has been associated with reduced bacterial biodiversity, most notably in moderate cases, and ecological divergence marked by enrichment of *Neisseria*, correlating with disease severity, alongside depletion of aerobic taxa like *Glaciecola* and facultative anaerobes like *Halomonas* [12]. In contrast with previous findings of OSA-related anaerobe abundance [69], obligate anaerobic taxa like *Anaerovorax* have been found to be diminished in severe OSA [12]. These shifts suggest a broad disruption of aerobic–anaerobic homeostasis, where microenvironmental alterations potentially driven by oxidative stress or hypoxia, favor the proliferation of pathobionts like *Neisseria* while displacing stability-associated taxa [12]. In the lower airways, bronchoalveolar lavage fluid analyses revealed phylum-level dysbiosis in OSA patients compared to healthy controls, with *Proteobacteria* and *Fusobacteria* dominating OSA-associated microbiota, while *Firmicutes* were significantly reduced. These findings were associated with nocturnal hypoxia and airway inflammation [80]. This distinct lung microbial signature in OSA patients also differed from those observed in other pulmonary diseases, suggesting OSA-specific alterations potentially driven by microaspiration of dysbiotic upper airway flora or impaired microbial clearance mechanisms intrinsic to OSA pathophysiology [81]. This study represents a key piece of direct evidence linking OSA to compositional shifts within the lung microbiome itself. While these findings highlight OSA as a disorder of microbiome-environment disequilibrium, limitations such as small sample sizes and heterogeneous control groups constrain generalizability. Further research is needed to disentangle causal relationships between microbial shifts, hypoxia and inflammation, and to assess whether these profiles represent unique biomarkers or therapeutic targets for OSA management.

### 3.2. Mechanistic Links Between the Lung Microbiome and OSA

An overview of proposed mechanisms is illustrated in Figure 2.

#### 3.2.1. Intermittent Hypoxia and Systemic Inflammation

Sleep disorders involving circadian rhythm disruption and sleep-disordered breathing (SDB), such as OSA, demonstrate particularly strong connections to respiratory infections and inflammation. SDB induces a pro-inflammatory state through several mechanisms, including intermittent hypoxia, sleep fragmentation, increased oxidative stress and elevation of inflammatory mediators including tumor necrosis factor α (TNF-α), IL-6 and C-reactive protein (CRP), the concentrations of which have been correlated with SDB severity, particularly in individuals with OSA [82]. Reciprocal relationships may further exacerbate upper airway dysfunction as IH-induced inflammatory processes in the carotid body have been shown to perturb immunoregulation and respiratory control, predisposing to nocturnal respiratory instability [83,84,85]. Additionally, TNF-α elevations, as observed in obesity, may impair upper airway musculature and increase collapse susceptibility [86].

OSA, characterized by recurrent episodes of IH and sleep fragmentation, drives systemic inflammation and oxidative stress, with profound implications for respiratory health and microbial homeostasis. OSA-induced repetitive cycles of IH are characterized by oscillations in oxygen saturation that have been shown to generate ROS via pathways analogous to ischemia–reperfusion injury [87], overwhelming endogenous antioxidant defenses, leading to oxidative stress and altered antioxidant enzyme activity and causing cellular damage [88]. It has been argued that this oxidative milieu is compounded by IH-mediated upregulation of pro-inflammatory cytokines, including TNF-α, IL-6 and IL-8, which exacerbate systemic inflammation and endothelial dysfunction, contributing to OSA comorbidities [82]. Notably, elevated TNF-α levels in OSA have been associated with excessive daytime sleepiness (EDS) [89] and genetic polymorphisms in the TNF-α gene (−308) cluster in OSA patients with EDS [90,91]. Pilot studies have suggested that pharmaceutical TNF-α inhibition (e.g., etanercept) reduces EDS, implicating TNF-α as both a biomarker and therapeutic target [92]. IL-6, a key mediator of OSA-associated morbidity, was found to be elevated in adipose tissue and plasma of obese OSA patients, with levels being attenuated by CPAP therapy [93]. IL-6 has also been correlated with EDS severity [94], though significant inter-individual heterogeneity existed in adults and children, likely influenced by genetic and environmental factors [95]. While this may be an adaptive response to hypoxemia at first, continuous triggering of the mechanism could collectively disrupt the lung microenvironment [96], altering pH, nutrient availability and epithelial barrier integrity—conditions that favor dysbiosis by shifting microbial fitness and host-microbe interactions [97,98]. This relationship perhaps helps explain why OSA patients face increased risks for respiratory infections and associated complications, due to hypoxia-driven selection pressures within the microbiome, creating another connection point between sleep disruption and respiratory inflammation.

Notably, severe OSA has been associated with increased oral and airway microbial diversity, including enrichment of pathobionts such as *Rothia*, *Haemophilus* and *Actinomyces*, likely due to hypoxia-induced shifts in bacterial fitness, host-microbe interactions [97,98,99] and innate defenses impairment, that reduce mucociliary clearance and alveolar macrophage efficacy, thereby facilitating pathogenic colonization [97,98]. Critically, the LRT is exposed to the same intermittent hypoxia and inflammatory mediators driving these upper airway shifts. It has been demonstrated that pro-inflammatory cytokines like IL-6 and IL-8 are elevated in pharyngeal lavage (PHAL) of OSA patients, reflecting mucosal inflammation and histological changes like connective tissue deposition that could increase airway collapsibility [100]. Given the continuum of the respiratory mucosa [37,38] and the established impact of such cytokines on epithelial barrier function and immune cell activity throughout the airways [5,53,54], it is highly plausible that this inflammatory milieu similarly disrupts the ecological balance and host defense mechanisms within the lung microenvironment, fostering LRT dysbiosis as directly observed by [80]. Critically, a reduction in PHAL inflammation [101,102] and cytokine levels [103] followed CPAP therapy. This reversibility underscores IH as a central driver of OSA-related pathophysiology, highlighting the potential for early intervention to mitigate microbial and immunological consequences.

#### 3.2.2. Mechanical Stress and Aspiration

The upper airway in OSA is prone to collapse due to passive airway narrowing and insufficient compensatory neural drive to compliant upper airway muscles during sleep [104]. This collapse has been shown to generate mechanical stress through repeated airway closure and reopening, inducing mucosal inflammation, subepithelial edema [105,106] and elevated inflammatory biomarkers like exhaled nitric oxide [106,107]. While such inflammation may alter the local microenvironment, potentially disrupting the lung microbiome, another documented factor of LRT dysbiosis seems to be the exacerbated aspiration risk due to apnea-related airway instability. Incomplete swallowing, linked to caudal hyoid displacement in OSA, has been correlated with delays in laryngeal elevation, leaving pharyngeal residues that provoke uncoordinated deglutition and microaspiration [35]. Aspiration introduces oral and gut microbes into the lungs, which may perturb the lung microbiome—a community of low biomass but high functional relevance, as evidenced by the high prevalence of recurrent pneumonia due to sleep-related aspiration, especially in older adults, where unwitnessed nocturnal aspiration is common [108]. This microaspiration of a perturbed microbial community is a direct mechanism by which upper airway OSA dysbiosis can seed and disrupt the lung microbiome, as supported by the distinct BALF profiles found in OSA [80,81]. Furthermore, CPAP therapy has been documented to mitigate this risk by stabilizing the airway, reducing uncoordinated swallowing (inspiration-associated deglutition) and promoting expiration-synchronized swallows, thereby lowering aspiration frequency [109], with direct evidence linking CPAP to lung microbiome normalization, though remaining unexplored, representing a critical research gap.

While the proposed mechanisms are plausible, causal evidence linking OSA to lung microbiome disruption is limited. Current evidence leans toward IH as the primary instigator of lung microbiome remodeling, given CPAP’s restorative effects, yet most studies are observational, correlating OSA with inflammation or aspiration events rather than microbiome changes. CPAP therapy addresses these pathways by improving airway stability and deglutition coordination, yet its microbiome-specific impacts are unproven and longitudinal studies profiling the lung microbiome pre- and post-CPAP, alongside mechanistic animal models, are needed.

#### 3.2.3. Could Baseline Lung Microbiome Composition Increase OSA Risk or Severity?

While hypoxia and inflammation driven by OSA are established contributors to microbial imbalance in the airways, recent evidence posits that pre-existing dysbiosis may itself prime the upper airway for collapsibility and inflammatory dysregulation, fostering a self-perpetuating cycle. This interplay is further reinforced by clinical correlations between elevated inflammatory biomarkers, OSA severity (as measured by AHI) and disrupted microbial communities, as inflammation appears to perpetuate dysbiosis, while dysbiosis amplifies pro-inflammatory signaling pathways, suggesting a reciprocal feedback loop [97]. These findings underscore the dual role of the lung microbiome as both a potential biomarker for OSA progression and a therapeutic target to disrupt this pathogenic cycle.

Composition of the respiratory tract has been implicated in the pathophysiology of chronic inflammatory diseases like asthma and COPD [6,7,110], comorbidities frequently intertwined with OSA [111]. Notably, perturbations in the relative abundance of a select subset of respiratory tract bacteria have been consistently linked to diverse chronic inflammatory conditions, with a recurring theme across studies highlighting the central role of IL-17 signaling pathways in mediating host responses to microbial community disturbances [112]. Perturbations in airway microbial communities such as elevations in *Haemophilus*, *Moraxella*, *Streptococcus* and *Neisseria* taxa, alongside reductions in *Veillonella* and *Prevotella*, have been linked to neutrophilic inflammation and IL-17-driven tissue remodeling [110,113,114,115] and may synergistically contribute to OSA pathophysiology. Elevated circulating inflammatory chemokines in OSA patients have been linked to neuromuscular activity impairment, potentially diminishing compensatory responses to upper airway obstruction during sleep [100]. Concurrently, mucosal inflammation may attenuate local afferent signaling and neuromuscular reflexes, exacerbating airway collapse [100]. Regarding the oropharynx, bacterial dysbiosis has been underscored as a hallmark of disease progression, with *Neisseria* abundance escalating with OSA severity, *Glaciecola* being absent in moderate/severe OSA, and *Tannerella*, *Anaerovorax* and *Halomonas* being depleted in severe cases [12]. The absence of *Glaciecola* in moderate/severe OSA could signify loss of oxidative stress buffering capacity, potentially exacerbating mucosal inflammation and conversely, the marked enrichment of Neisseria—a genus linked to mucosal colonization and immune modulation—could suggest its role in destabilizing local immunity through endotoxin production [12]. Such dysbiosis may perpetuate a feed-forward cycle of inflammation and airway dysfunction, mirroring mechanisms observed in asthma, where low microbial diversity have been correlated with airflow obstruction [116,117]. Indeed, elevated nasopharyngeal levels of anaerobic bacteria and dominant phyla have been inversely correlated with nocturnal oxygen saturation nadir [69] while nasal lavage enrichment of *Streptococcus*, *Prevotella* and *Veillonella* aligned with AHI values and elevated cytokine levels [68]. Furthermore, CPAP therapy demonstrated no significant alterations in nasal microbiota composition, suggesting that microbiome perturbations linked OSA may exhibit stability and resistance to short-term resolution via mechanical airway stabilization [68]. However, this observation, derived from a singular study with limited sample size, requires cautious interpretation, as broader validation and longitudinal analyses are necessary to confirm the temporal dynamics of microbiome-host interactions in OSA. These findings could highlight OSA as a disorder of microbial-immune crosstalk, wherein hypoxia and inflammation drive ecological collapse, favoring pathobionts like *Neisseria* while depleting protective taxa. Further research is warranted to delineate causal pathways linking dysbiosis, inflammation and neuromuscular impairment in OSA progression.

Finally, it is worth noting that early-life LRT infections have been associated with later OSA diagnosis, implicating developmental microbial-immune crosstalk in upper airway neuromotor control [118]. Neonatal colonization by *Streptococcus pneumoniae* or *Moraxella catarrhalis* increased asthma susceptibility and also enhanced β2-agonist reversibility, reflecting airway hyperreactivity [119,120]—a trait shared with OSA. These findings suggest that early microbial disturbances may program immune responses toward inflammatory phenotypes, increasing vulnerability to airway collapse during sleep [119].

## 4. Limitations

A particularly significant limitation of this review is presented by the current diagnostic hurdles stemming from the lung microbiome’s low biomass, the contamination risks during sampling and the methodological variability across studies. Traditional culture-dependent techniques often overlook low-abundance taxa critical to disease pathogenesis, while bronchoscopic sampling introduces upper respiratory tract contaminants. Advances in metagenomic sequencing, metabolomics and single-cell RNA profiling offer promise for resolving these limitations, enabling precise identification of microbial signatures and host-microbe interactions. Non-invasive biomarkers such as exhaled volatile organic compounds could revolutionize OSA diagnostics by capturing real-time dysbiosis without invasive procedures. Integrating multi-omics data with machine learning may further distill microbiome-host interplay into predictive models for disease stratification and therapeutic response. Finally, microbial signatures could serve as biomarkers for OSA severity, predict CPAP responsiveness or identify candidates for microbiota-targeted therapies. While CPAP therapy partially restores microbial equilibrium, highlighting hypoxia as a central driver of dysbiosis, efficacy varies with baseline microbiome composition and genetic factors. Hence, integrating microbiome data with polysomnographic, genetic and metabolic profiles will enable holistic patient stratification, transforming OSA from a mechanical disorder into a treatable ecosystem imbalance, though rigorous safety and efficacy data are needed.

## 5. Conclusions and Future Research Directions

The complex relationship between microbiome-induced respiratory inflammation and sleep disorders represents an emerging research frontier with significant clinical relevance. The evidence examined in this narrative review underscores a dynamic interplay between the lung microbiome and OSA that connect through shared inflammatory pathways, oxidative stress mechanisms and systemic immune responses, creating potential feedback loops that can exacerbate both respiratory dysfunction and sleep disturbances through sustained inflammatory signaling. OSA-associated IH, mechanical stress from airway collapse and systemic inflammation drive dysbiotic shifts in respiratory microbial communities, favoring anaerobic taxa and depleting aerobic symbionts, while elevated pro-inflammatory cytokines and oxidative stress impair mucosal immunity, reduce microbial clearance and perpetuate airway instability. Conversely, dysbiosis itself may prime the upper airway for collapsibility by amplifying inflammation and disrupting neuromuscular reflexes, thus creating a self-reinforcing cycle.

Interconnections between microbiome-induced respiratory inflammation and sleep disorders, present significant clinical implications. These relationships suggest novel approaches to diagnosis, monitoring and treatment that consider the complex interplay between these systems rather than addressing each component in isolation. The elevation in inflammatory markers does not only reflect underlying disease processes, but may actively contribute to symptom development and progression, making them valuable diagnostic tools, as identifying specific inflammatory markers and oxidative stress indicators associated with both microbiome dysbiosis and sleep disturbances could improve risk assessment and enable earlier intervention. Oxidative stress biomarkers provide insights into the biochemical imbalances mediating the relationship between these conditions, offering additional intervention opportunities through antioxidant approaches, as research indicates a positive correlation between elevated oxidative balance scores and improved sleep quality, suggesting that antioxidant-rich diets and healthy lifestyle choices may help address sleep-related concerns [121]. This approach could potentially benefit respiratory health simultaneously, given oxidative stress’s role in respiratory inflammation and microbiome regulation. Moreover, alongside elevated cytokine levels, they might further help in identifying individuals at risk for developing both respiratory inflammation and sleep disorders before clinical symptoms fully manifest. Additionally, longitudinal monitoring of these markers could provide insights into disease progression and treatment response, potentially enabling more personalized therapeutic strategies based on individual inflammatory profiles. Future research should focus on several key directions to advance understanding of these complex relationships. Longitudinal studies examining temporal relationships between microbiome alterations, respiratory inflammation markers and sleep parameters would clarify whether these conditions develop simultaneously or follow particular sequences.

## Figures and Tables

**Figure 1 diagnostics-15-01431-f001:**
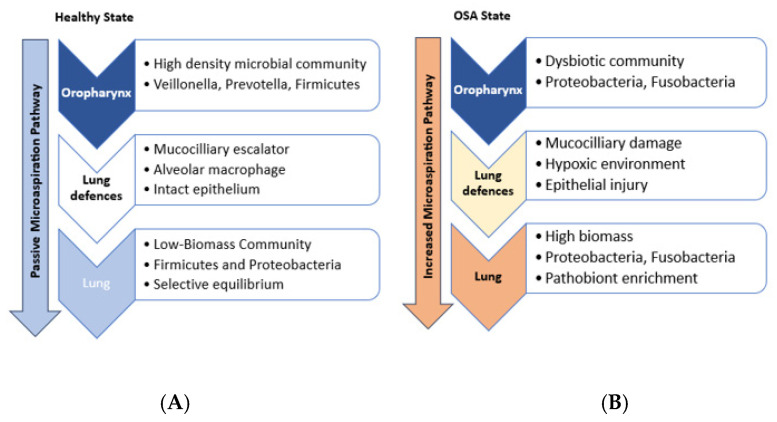
Oropharyngeal-Lung Microbiome Dynamics in Health and Obstructive Sleep Apnea (OSA). (**A**) Healthy State: The lung microbiome is seeded primarily via passive microaspiration of oropharyngeal microbiota. Robust lung defenses maintain a low-biomass, selectively filtered community through equilibrium between microbial immigration and elimination. (**B**) OSA state: OSA pathophysiology (intermittent hypoxia, inflammation, mechanical stress) induces oropharyngeal dysbiosis and increases microaspiration volume. Concurrently, OSA impairs lung defenses and creates a hypoxic microenvironment. This disrupts the immigration–elimination equilibrium, permitting dysbiotic expansion of pathobionts like Proteobacteria and Fusobacteria in the lower respiratory tract.

**Figure 2 diagnostics-15-01431-f002:**
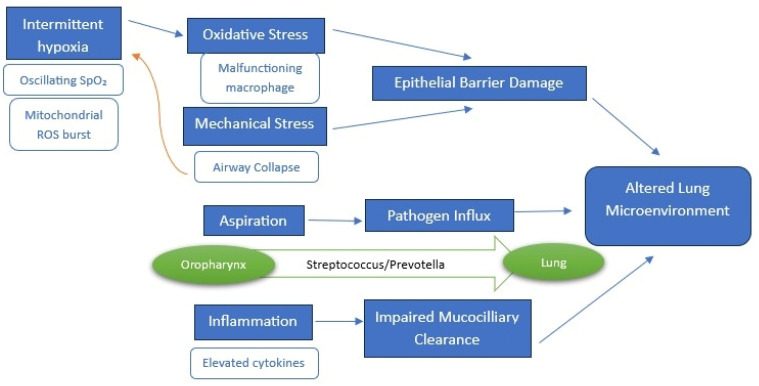
Pathogenic cycle of OSA-driven lung microbiome dysbiosis. OSA pathophysiological drivers (intermittent hypoxia, mechanical stress, systemic inflammation, aspiration) disrupt lung homeostasis. This alters the pulmonary microenvironment through oxidative damage, impaired clearance and pathogen influx. The airway collapse -a hallmark of OSA- further perpetuates intermittent hypoxia and the cascade it triggers. Resulting dysbiosis (Proteobacteria/Fusobacteria enrichment, Firmicutes depletion) exacerbates OSA via endotoxin-mediated inflammation and impaired mucociliary completing a self-perpetuating cycle.

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
