# Peer review of "The Lung Microbiome and Its Impact on Obstructive Sleep Apnea: A Diagnostic Frontier"

_diagnostics, 2025, doi:10.3390/diagnostics15111431_

Round 1

Reviewer 1 Report

Comments and Suggestions for Authors

The manuscript `The Lung Microbiome and Its Impact on Obstructive Sleep Apnea: A Diagnostic Frontier´ by Aliki Karkala et al summarized current knowledge on the bidirectional interplay between obstructive sleep apnea and lung microbial communities and tried to highlight how hypoxia-induced alterations in microbial ecology disrupt immune homeostasis. 
The review topic is interesting, but the review needs to be improved on the focus of the statements. The aim/objective should be more focused.
Authors should correct grammatical errors. 

Author Response

Reviewer 1 comments: The review topic is interesting, but the review needs to be improved on the focus of the statements. The aim/objective should be more focused.
Authors should correct grammatical errors. 

Answer: Thank you for your comments. The objective of the study has been rewritten as follows (and highlighted in the text): Specifically, it aims to: 1. examine how core OSA pathophysiological features such as intermittent hypoxia, mechanical stress and systemic inflammation drive compositional and functional alterations in the lung microbiome; 2. explore the mechanisms by which resultant dysbiosis may perpetuate or exacerbate OSA severity through inflammatory cascades, immune dysregulation and impaired airway stability; 3. evaluate the potential of specific microbial signatures and associated inflammatory markers as biomarkers for OSA severity, progression, or therapeutic response. By integrating insights from microbial ecology, immunology and sleep medicine, this review seeks to advance the understanding of OSA as a disorder of disrupted host-microbe crosstalk and highlight the lung microbiome's emerging role as a diagnostic and therapeutic target.

Furthermore, grammatical errors have been corrected and highlighted throughout the text.

Reviewer 2 Report

Comments and Suggestions for Authors

Dear Editor,

The authors review the current knowledge on the bidirectional interaction between obstructive sleep apnea (OSA) and lung microbial communities. The complex relationship between microbiome-induced airway inflammation and sleep disorders is an emerging topic. The manuscript is well written and a detailed literature review has been carried out.

Sincerely

Comments on the Quality of English Language

The English could be improved to more clearly express the research.

Author Response

Comments from Reviewer 2: The authors review the current knowledge on the bidirectional interaction between obstructive sleep apnea (OSA) and lung microbial communities. The complex relationship between microbiome-induced airway inflammation and sleep disorders is an emerging topic. The manuscript is well written and a detailed literature review has been carried out.

Responce: Thank you very much for your comments.

Reviewer 3 Report

Comments and Suggestions for Authors

Although the proposed topic is interesting, I believe there is not enough information, or at least the information presented by the authors fails to convey the link or importance between pulmonary microbiome and obstructive sleep apnea, since much of the referenced work primarily focuses on the effect of microbiome from other anatomical sites.
1. In section 2, which discusses the composition of the pulmonary microbiome, it remains unclear how LRT is influenced by oropharyngeal biome, even though they are anatomically located in separate sites. This topic should be discussed, and it is advisable to add a figure that schematizes the probable mechanism (proposed by the authors) involved.
2. Line 104 mentions that fungi act as cofactors of immune modulation. What are the authors referring to?
3. In section 3: Microbiome Dysbiosis and OSA, discussing microbial profiles (lines 281-303), it is unclear whether the cited disruptions are in the lung biome or elsewhere. If not in the lung biome, how do the authors intend to relate it to the lung and OSA?
4. In the following section, Mechanistic Links Between the Lung Microbiome and OSA, the authors are encouraged to outline the proposed mechanism in a second figure.
5. Considering that lifestyle and diet greatly influence the gut microbiota, would something similar happen with the lung biota? In this regard, have any patterns of higher prevalence of obstructive sleep apnea been reported due to a certain type of diet or geographic location? Discuss.

Comments on the Quality of English Language

The English should be thoroughly reviewed as some ideas are confusing in the text.

Author Response

Comment 1: Although the proposed topic is interesting, I believe there is not enough information, or at least the information presented by the authors fails to convey the link or importance between pulmonary microbiome and obstructive sleep apnea, since much of the referenced work primarily focuses on the effect of microbiome from other anatomical sites.

Answer 1: Thank you for your insight. Additional information can be found:

  1. In the subsection “Emerging Research on Microbial Profiles in OSA Patients”, highlighted in the text and as follows: “primarily due to the invasiveness and challenges of sampling the low-biomass LRT. Consequently, most available data derive from studies of the upper airways (nasopharynx, oropharynx), which, as the primary source of lung microbes via microaspiration [1,35,36], provide crucial, mechanistically relevant proxies for inferring LRT alterations, given this anatomical and physiological continuum.” & “This distinct lung microbial signature in OSA patients also differed from those observed in other pulmonary diseases, suggesting OSA-specific alterations potentially driven by microaspiration of dysbiotic upper airway flora or impaired microbial clearance mechanisms intrinsic to OSA pathophysiology [81]. This study represents a key piece of direct evidence linking OSA to compositional shifts within the lung microbiome itself.”
  2. In the subsection “Intermittent hypoxia and systemic inflammation”, highlighted in the text and as follows: “Critically, the LRT is exposed to the same intermittent hypoxia and inflammatory mediators driving these upper airway shifts.” & “Given the continuum of the respiratory mucosa [37,38] and the established impact of such cytokines on epithelial barrier function and immune cell activity throughout the airways [5,53,54], it is highly plausible that this inflammatory milieu similarly disrupts the ecological balance and host defense mechanisms within the lung microenvironment, fostering LRT dysbiosis as directly observed by [80].”
  3. In the subsection “Mechanical Stress and Aspiration”, highlighted in the text and as follows: “This microaspiration of a perturbed microbial community is a direct mechanism by which upper airway OSA dysbiosis can seed and disrupt the lung microbiome, as supported by the distinct BALF profiles found in OSA [80, 81].”

Comment 2: In section 2, which discusses the composition of the pulmonary microbiome, it remains unclear how LRT is influenced by oropharyngeal biome, even though they are anatomically located in separate sites. This topic should be discussed, and it is advisable to add a figure that schematizes the probable mechanism (proposed by the authors) involved.

Answer 2: Thank you for your comment. The revised section “Potential origin of lung microbiome” addresses this request as follows (and highlighted in the text): “Evidence has suggested that the lung microbiome may originate from the URT, with the primary mechanism linking these anatomically contiguous sites being microaspiration [1,35,36]. During normal respiration and particularly during sleep, small volumes of oropharyngeal secretions are passively aspirated into the LRT (see Figure 1). This process introduces a continuous influx of microbial immigrants from the high-biomass oro-pharynx into the low-biomass lung environment [1,35].” & “This combination of immigration and elimination, with minimal local replication under healthy conditions, shapes a transient, low-biomass community distinct from its oropharyngeal source. The selective pressures exerted by lung-specific clearance mechanisms and the physico-chemical environment mean the LRT microbiome is not simply a diluted copy of the oropharynx but a selectively filtered derivative [3,14,39,40]. However, disease states disrupt this equilibrium, and structural damage, impaired clearance through reduced mucociliary function, inflammation, or altered conditions like hypoxia in OSA, can favor microbial proliferation, overriding the immigration-clearance balance and allowing dysbiotic communities to establish [1].” & “Consequently, OSA may increase both the volume and altered composition of aspirated material while simultaneously reducing the LRT's capacity to clear or control it, creating a perfect storm for seeding and perpetuating lung dysbiosis (Figure 1). This establishes a direct pathophysiological pathway linking OSA to potential LRT microbial alterations via its impact on the oropharynx and clearance mechanisms.”. A figure has also been provided.

Comment 3: Line 104 mentions that fungi act as cofactors of immune modulation. What are the authors referring to?

Answer 3: Thank you for this comment. The following text has been added and has been highlighted in the manuscript: Unlike bacteria, which often directly influence pulmonary homeostasis through active colonization and metabolite production, current evidence suggests fungi primarily function as immunomodulatory cofactors. This implies they do not typically initiate or dominantly regulate baseline pulmonary immune tone themselves. Instead, they modulate ongoing immune responses potentially initiated by bacteria, viruses, or other stimuli, through interactions with host pattern recognition receptors and the release of metabolites that can amplify or dampen inflammatory pathways [22].

Comment 4
: In section 3: Microbiome Dysbiosis and OSA, discussing microbial profiles (lines 281-303), it is unclear whether the cited disruptions are in the lung biome or elsewhere. If not in the lung biome, how do the authors intend to relate it to the lung and OSA?

Answer 4: Thank you for your insightful comment. The authors believe that this has been answered in “Answer 1”.

Comment 5: In the following section, Mechanistic Links Between the Lung Microbiome and OSA, the authors are encouraged to outline the proposed mechanism in a second figure.

Answer 5:  Thank you for this comment. A second figure has been provided.

Comment 6: Considering that lifestyle and diet greatly influence the gut microbiota, would something similar happen with the lung biota? In this regard, have any patterns of higher prevalence of obstructive sleep apnea been reported due to a certain type of diet or geographic location? Discuss.

Answer 6: Thank you for this interesting comment. After a search of the literature, the authors found that several studies discuss microbiota changes in adjacent sites, which may influence lung ecology through systemic pathways. Current studies however, focus on gut microbiota, perhaps due to sampling challenges in the low-biomass lung environment. While lifestyle and dietary patterns have been found to influence OSA risk through effects on obesity and inflammation, the authors did not find any studies on higher prevalence of OSA due to lung microbiome shifts in these patients. Similarly, regional microbiome variations (e.g. higher Prevotella in agrarian diets vs. Bacteroides in industrialized nations) are underexplored in OSA contexts.

Finally, the text has been reviewed for grammatical errors.

Reviewer 4 Report

Comments and Suggestions for Authors

Comments to the Authors: Your efforts are appreciated.

Reference number: diagnostics-3637878

Title: The Lung Microbiome and Its Impact on Obstructive Sleep Apnea: A Diagnostic Frontier.

In order to evaluate the state of knowledge on the reciprocal interactions between lung microbial communities, which include bacteria, viruses, and fungi, and obstructive sleep apnea (OSA), the authors attempted to analyze the data from the literature.  The study demonstrated how inflammation-driven mucosal damage promotes pathogenic colonization, while hypoxia-induced changes in microbial ecology upset immunological homeostasis.  Clinical associations between the severity of the condition and certain microorganisms, point to microbial signatures as new biomarkers for the development of OSA and the effectiveness of treatment.  Additionally, pro-inflammatory cytokines and indicators of oxidative stress show promise as markers that link microbial dysbiosis to sleep-related clinical conditions.  The information in the manuscript may help future studies concentrate on many important areas to improve our comprehension of these intricate relationships.

Overall comments

* The manuscript well-written, comprehensive, and offers useful information about the impact of the lung microbiome and dysbiosis on the progression of OSA.

*Overall, the manuscript is interesting because it perfectly highlighted the composition of the lung microbiome in healthy individuals (including bacteria, viruses, and fungi), origin of lung microbiome, pathological dysbiosis and its implications for respiratory inflammation and immune response, and role of microbiome and dysbiosis in OSA. According to the data in this review, the lung microbiota and OSA interact dynamically through shared inflammatory pathways, oxidative stress mechanisms, and systemic immune responses. This interaction may result in feedback loops that worsen respiratory dysfunction and sleep disturbances by causing persistent inflammatory signaling.  While increased pro-inflammatory cytokines and oxidative stress weaken mucosal immunity, lower microbial clearance, and prolong airway instability, OSA-associated intermittent hypoxia, mechanical stress from airway collapse, and systemic inflammation cause dysbiotic shifts in respiratory microbial communities that favor anaerobic and deplete aerobic bacteria. Thus, the manuscript is acceptable.

*The English editing of the study: Very good.

*The similarity index of the manuscript text: Very good (13%).

*Names of the organisms such as “Streptococcus pneumoniae in line number 203, 456, and Moraxella catarrhalis in line number 457”: The name should be in italic throughout the manuscript if using the genus and species.

Specific comments

Title: Clearly reflects the content of the manuscript.

Keywords: keywords are representative of the research. They should be arranged alphabetically.

Abstract: Sufficiently reflects the content of the manuscript.

Introduction: Provides sufficient background.

The aim of the study: Clearly described.

The literature review: Informative, and in line with the aim of the study.

Limitation of the study: Kindly, should be shifted to be before the conclusions and future research directions.

Conclusions and future research directions: Informative, and consistent with the evidence presented.

References:

  • Relevant, and most are recent.
  • Match the reference style of the journal.

Thanks

Author Response

Comments of Reviewer 4: 

Comments to the Authors: Your efforts are appreciated.

Reference number: diagnostics-3637878

Title: The Lung Microbiome and Its Impact on Obstructive Sleep Apnea: A Diagnostic Frontier.

In order to evaluate the state of knowledge on the reciprocal interactions between lung microbial communities, which include bacteria, viruses, and fungi, and obstructive sleep apnea (OSA), the authors attempted to analyze the data from the literature.  The study demonstrated how inflammation-driven mucosal damage promotes pathogenic colonization, while hypoxia-induced changes in microbial ecology upset immunological homeostasis.  Clinical associations between the severity of the condition and certain microorganisms, point to microbial signatures as new biomarkers for the development of OSA and the effectiveness of treatment.  Additionally, pro-inflammatory cytokines and indicators of oxidative stress show promise as markers that link microbial dysbiosis to sleep-related clinical conditions.  The information in the manuscript may help future studies concentrate on many important areas to improve our comprehension of these intricate relationships.

Overall comments

* The manuscript well-written, comprehensive, and offers useful information about the impact of the lung microbiome and dysbiosis on the progression of OSA.

*Overall, the manuscript is interesting because it perfectly highlighted the composition of the lung microbiome in healthy individuals (including bacteria, viruses, and fungi), origin of lung microbiome, pathological dysbiosis and its implications for respiratory inflammation and immune response, and role of microbiome and dysbiosis in OSA. According to the data in this review, the lung microbiota and OSA interact dynamically through shared inflammatory pathways, oxidative stress mechanisms, and systemic immune responses. This interaction may result in feedback loops that worsen respiratory dysfunction and sleep disturbances by causing persistent inflammatory signaling.  While increased pro-inflammatory cytokines and oxidative stress weaken mucosal immunity, lower microbial clearance, and prolong airway instability, OSA-associated intermittent hypoxia, mechanical stress from airway collapse, and systemic inflammation cause dysbiotic shifts in respiratory microbial communities that favor anaerobic and deplete aerobic bacteria. Thus, the manuscript is acceptable.

*The English editing of the study: Very good.

*The similarity index of the manuscript text: Very good (13%).

*Names of the organisms such as “Streptococcus pneumoniae in line number 203, 456, and Moraxella catarrhalis in line number 457”: The name should be in italic throughout the manuscript if using the genus and species.

Answer to general comments: Thank you very much for your comments. Names of microorganisms have been edited in italic throughout the manuscript. The changes are highlighted in the text.

Specific comments

Title: Clearly reflects the content of the manuscript.

Keywords: keywords are representative of the research. They should be arranged alphabetically.

Abstract: Sufficiently reflects the content of the manuscript.

Introduction: Provides sufficient background.

The aim of the study: Clearly described.

The literature review: Informative, and in line with the aim of the study.

Limitation of the study: Kindly, should be shifted to be before the conclusions and future research directions.

Conclusions and future research directions: Informative, and consistent with the evidence presented.

References:

  • Relevant, and most are recent.
  • Match the reference style of the journal.

Answer to specific comments: Thank you for your input. Keywords have been arranged alphabetically, limitations have been shifted to appear before conclusions and references are in IEEE form (sources sited through "Mendeley Cite" reference manager).

Round 2

Reviewer 3 Report

Comments and Suggestions for Authors

The document had considerable improvements, and each of the comments were properly addressed.